# Barriers and Facilitators to Smoking Cessation Among University Students: A Scoping Review

**DOI:** 10.3390/ijerph22060947

**Published:** 2025-06-17

**Authors:** Farhan Alanazi, Walid Jumaa Mohamed Mohamed, Stathis Th. Konstantinidis, Holly Blake

**Affiliations:** 1School of Health Sciences, University of Nottingham, Nottingham NG7 2HA, UK; ntxfa22@nottingham.ac.uk (F.A.); walid.mohamed3@nottingham.ac.uk (W.J.M.M.); stathis.konstantinidis@nottingham.ac.uk (S.T.K.); 2Health Informatics Department, School of Health Sciences, Saudi Electronic University, Jeddah 11673, Saudi Arabia; 3NIHR Nottingham Biomedical Research Centre, Nottingham NG7 2UH, UK

**Keywords:** public health, health promotion, university students, smoking cessation, scoping review

## Abstract

University students are a vulnerable population for smoking initiation and continuation, often facing unique challenges in accessing cessation support. This scoping review aimed to map the existing literature on the barriers and facilitators to smoking cessation among university students using the Theoretical Domains Framework (TDF). Following the JBI methodology, six studies conducted in the United States, Jordan, and Qatar were included, employing both qualitative and quantitative designs. A total of 22 barriers and 20 facilitators were identified and mapped across relevant TDF domains. Key barriers included time constraints, financial limitations, low self-efficacy, and social smoking norms. Facilitators included access to flexible, low-cost interventions, peer support, and previous quit attempts. Digital interventions were preferred by students. The findings suggest that smoking cessation strategies targeting university students should be flexible, affordable, and embedded within campus health systems. Interventions that combine behavioral support, peer involvement, and accessible technology show strong potential in addressing the multifaceted barriers faced by this population. This review provides a theory-informed foundation for the development of tailored smoking cessation interventions and identifies key directions for future research.

## 1. Introduction

The prevalence of smoking remains a significant global health issue, with approximately 1.25 billion adults worldwide currently using tobacco [1]. Despite a decline in tobacco use rates since 2000, smoking continues to be a leading cause of preventable morbidity and mortality, contributing to serious health outcomes including cardiovascular diseases, cancer, and respiratory illnesses [1,2]. Economically, smoking imposes a substantial burden. In England, the total societal cost of smoking is estimated at GBP 43.7 billion annually [3]. In the United States, the total economic burden of smoking exceeded USD 600 billion in 2018 [4]. Globally, a recent estimate places the social and economic burden of tobacco use at approximately USD 1.7 trillion in 2022, equivalent to 1.7% of the annual global gross domestic product [5]. Among the different types of tobacco use, conventional cigarettes remain the most common form globally, associated with the greatest adverse health consequences [6]; therefore, cigarette smoking is the focus of this review. Here, the term “smoking cessation” refers specifically to cessation of cigarette smoking.

Smoking-related diseases resulted in approximately 8 million deaths worldwide in 2019 alone [7]. Certain populations are particularly vulnerable to smoking initiation and continuation. Among these, university students face unique risks due to factors such as greater independence, exposure to new social environments, and increased academic pressures [8,9,10]. Transition to university life is a critical developmental stage, during which students form their identities and develop personal beliefs and values [11], all of which may contribute to the formation of long-term health behaviors. Students often underestimate the risks associated with harmful behaviors such as smoking [12] and are frequently exposed to strong peer influences and the easy availability of cigarettes [13]. These factors contribute to their increased vulnerability to smoking. This vulnerability is reflected in studies documenting the high prevalence of smoking among university students in different geographical regions representing diverse cultural, economic, and tobacco-control contexts. In these studies, smoking prevalence among university students has been found to be 48.8% in Sudan [14], 32.6% in Bangladesh [15], 27.9% in Türkiye [16], 26% in France [17], 17.7% in England [18], and 7.8% to 17.5% in Saudi Arabia [19]. This high prevalence may lead to multiple consequences. In addition to the adverse health effects of smoking and its association with other risky behaviors such as alcohol abuse [8,20], smoking may also negatively affect students’ academic performance, contributing to higher rates of absenteeism and academic warnings [21]. Although various smoking cessation interventions have been developed for university students, the most effective approach to changing tobacco use behavior remains uncertain [22]. To develop effective smoking cessation interventions, it is crucial to understand the factors that influence smoking cessation among this population, identifying both the barriers they face and the facilitators that can support their success [23,24].

Several reviews have examined barriers and facilitators to smoking cessation across different populations but do not focus specifically on university students. For example, Twyman et al. investigated perceived barriers among various vulnerable groups [25]. Flemming et al. also reviewed qualitative research on smoking in pregnancy [26]. Zvolska et al. assessed the facilitators and barriers to smoking cessation among patients with tuberculosis in low- and middle-income countries [27]. In the context of mental health, Trainor and Leavey conducted a review of barriers and facilitators to smoking cessation among individuals with severe mental illness [28]. Lum et al. examined psychosocial barriers and facilitators to smoking cessation in people living with schizophrenia [29]. These reviews highlight the complexity of smoking cessation, considering personal, social, and structural factors that influence quitting. However, no studies have explicitly examine the unique experiences of university students, underscoring the need for a dedicated scoping review focused on this population.

To address this gap, this scoping review aims to map the existing literature on the barriers and facilitators to smoking cessation among university students. Scoping reviews are particularly suited for exploring broad questions across diverse methodologies and study designs [30,31]. A scoping review was chosen because of the diversity within university student populations, as well as the variety of contexts and study designs in this field. Unlike systematic reviews, which typically have a narrower focus and strict inclusion criteria, scoping reviews allow for broader exploration, helping to identify knowledge gaps and clarify key concepts [32,33]. This comprehensive mapping is an essential first step in the evidence synthesis process, aiming to identify the barriers and facilitators to smoking cessation among university students, along with the theoretical frameworks applied and study methodologies employed, thereby guiding later systematic reviews and more targeted research [34].

The review question was the following:

What are the barriers and facilitators of smoking cessation among university students?

## 2. Materials and Methods

A scoping review was conducted to explore the barriers and facilitators of smoking cessation among university students. This review followed the JBI methodology for scoping reviews [35]. This study design was appropriate, as the goal was to map and characterize the existing evidence rather than synthesize it in response to a specific research question [31]. The findings are reported in accordance with the Preferred Reporting Items for Systematic Reviews and Meta-Analyses extension for Scoping Reviews (PRISMA-ScR) guidelines [34] (Appendix A). The TDF was used to categorize the barriers and facilitators identified in the included studies [36]. It was selected as a comprehensive framework that consolidates constructs from multiple behavior change theories [37]. Its broad scope makes it especially well-suited to systematically identifying and organizing influences on smoking behavior, and it has been widely used in studies examining barriers and facilitators in the area of smoking cessation [38,39,40]. The review was registered with Open Science Framework on 10 February 2025: https://doi.org/10.17605/OSF.IO/NHK2Y (accessed on 13 June 2025).

### 2.1. Inclusion Criteria

The inclusion and exclusion criteria follow the participant, concept, and context (PCC) framework [41].

Participants: This review included studies focusing on university students. To meet the inclusion criteria, participants must be enrolled in undergraduate or graduate programs. Studies that included university students along with other populations, such as high school students, university staff, or healthcare professionals, were excluded to maintain relevance to the student experience.

Concept: This review included studies that primarily examined the barriers and facilitators to smoking cessation among university students. Studies were considered eligible if their primary focus was on identifying and analyzing the factors that hinder or support smoking cessation efforts. Studies that did not explicitly address barriers or facilitators to smoking cessation were excluded.

Context: The review focused on higher education settings, including universities, colleges, and similar institutions. Studies considering institutional, cultural, or social aspects relevant to university students were included. Studies conducted outside of these academic environments or those not specifically related to university students were excluded to ensure the findings remain applicable to the student experience within higher education settings.

### 2.2. Types of Sources

This scoping review adopted an inclusive approach to study design selection to comprehensively examine the barriers and facilitators influencing smoking cessation among university students. Quantitative studies, including cross-sectional, experimental, and quasi-experimental designs, were included to assess the prevalence and impact of specific barriers and facilitators. Qualitative studies were incorporated to provide an in-depth understanding of personal experiences, perceptions, and contextual factors affecting smoking cessation. Additionally, mixed-methods studies were included to integrate numerical data with qualitative insights, offering a holistic perspective on the factors influencing smoking cessation.

### 2.3. Search Strategy

The search strategy aimed to locate published studies relevant to the barriers and facilitators of smoking cessation among university students. A three-step approach was employed to ensure a comprehensive and systematic search. An initial limited search was conducted in MEDLINE (via Ovid) and CINAHL (via EBSCO) to identify relevant articles and extract keywords and index terms. Based on this preliminary search, a refined search strategy was developed in consultation with a librarian (Appendix A).

The finalized search terms were adapted and applied to multiple electronic databases, including MEDLINE (via Ovid), PsycInfo (via Ovid), Embase (via Ovid), CINAHL (via EBSCO), Web of Science, and Scopus. The reference lists of all included studies were screened for additional relevant sources to enhance the comprehensiveness of the review. The search strategy included studies published in English and Arabic, with no restrictions on publication date or geographical location, to ensure an inclusive and up-to-date overview of the available evidence.

### 2.4. Study/Source of Evidence Selection

Following the search, all identified citations were collated and uploaded into EndNote 21.5 (Build 18513) (Clarivate Analytics, Philadelphia, PA, USA) for management and deduplication. A pilot test was conducted to ensure consistency in applying the eligibility criteria. One reviewer (FA) screened the titles and abstracts of all citations, with independent verification by a second reviewer (W.J.M.M.) to enhance accuracy and minimize bias.

Potentially relevant citations were retrieved in full text and imported into Covidence (Veritas Health Innovation, Melbourne, Australia) for further assessment. The full text of these citations was evaluated against the inclusion criteria by one reviewer (FA), with independent verification by a second reviewer (W.J.M.M.). Reasons for excluding full-text sources that did not meet the inclusion criteria were recorded and reported in the final scoping review to ensure transparency.

The search and study selection process was fully documented in the final scoping review and visually presented using a Preferred Reporting Items for Systematic Reviews and Meta-Analyses Extension for Scoping Reviews (PRISMA-ScR) flow diagram [34].

### 2.5. Data Extraction

Data were extracted from the included studies by one reviewer (FA) using a data extraction tool adapted from the JBI guidance for scoping reviews [41]. The tool captured key study characteristics, including author(s), year of publication, country, study type, study design, data collection method, study aim, population, setting, smoking behavior, study sample, and key findings (Appendix A).

The tool was modified during the pilot phase, and all modifications are detailed within the review. Disagreements were resolved through discussion among the authors. The extracted data were then analyzed by the first and second reviewers (FA and W.J.M.M.).

The initial plan included contacting the authors of the respective studies to see if any data were missing or required further clarification. However, this was not necessary, as the extracted data were sufficient to address the research objectives.

### 2.6. Data Analysis and Presentation

This scoping review used the TDF to identify barriers and facilitators to smoking cessation [36]. The findings are presented in tables and a narrative summary to address the review question. The tables provide a structured overview of the data, while the narrative summary explains how the results relate to the review question.

## 3. Results

### 3.1. Study Inclusion

The search identified a total of 1313 records from the databases. After removing 757 duplicates, 556 studies were screened based on title and abstract. During this screening phase, 397 records were excluded for not being relevant to the review topic, 100 records were excluded as secondary studies, three records were study protocols without results, and 18 records were in languages other than English or Arabic.

Following this initial screening, 38 studies were assessed for full-text eligibility. Of these, 29 studies were excluded for not focusing on the barriers and facilitators of smoking cessation, two studies were excluded due to the wrong population, and one study was excluded for being in the wrong language.

Finally, six studies met the inclusion criteria and were included in the review. The study selection process is visually presented in Figure 1. A list of excluded studies with reasons can be found in Appendix A.

### 3.2. Characteristics of Included Studies

The studies included in this review provide insights into smoking behavior and factors influencing smoking cessation among university students across different countries. A total of six studies were identified, conducted in the United States [43,44,45,46], Jordan [47], and Qatar [48].

Various study designs were employed, including qualitative focus groups [43], cross-sectional studies [44,46,47,48], and descriptive studies [45], to explore barriers and facilitators to smoking cessation. Data collection methods included self-administered surveys [44,46,47], semi-structured interviews [48], and focus group discussions [43,45].

The sample sizes for the included studies varied widely, ranging from 19 to 4438 participants. Participants were predominantly university students aged between 18 and 30, with varying gender distributions.

Study settings reflected diverse university environments. Studies in the United States were conducted at large public universities and multi-campus settings [43,44,45,46]. Research in Jordan and Qatar focused on public university settings [47,48]. The characteristics of the included studies are presented in Table 1.

#### 3.2.1. Identified Barriers and Facilitators

Our review identified 22 barriers and 20 facilitators to smoking cessation, categorized according to the TDF. The reported domains included Knowledge [44,45], Skills [44,45,47], Social/Professional Role and Identity [44,47,48], Beliefs about Capabilities [46,47,48], Beliefs about Consequences [44,45,47,48], Intentions and Goals [43,44,47], Reinforcement [43,47], Environmental Context and Resources [43,45,46,48], Social Influences [43,44,45,47], and Emotion [44,45,47].

#### 3.2.2. Environmental Context and Resources

Barriers in this domain included time constraints due to academic schedules, which prevented students from accessing cessation services [48]. Financial constraints were also highlighted as a significant barrier, with students reporting difficulties affording nicotine replacement therapies [45]. Additionally, being in social environments with other smokers increased the likelihood of relapse, making it difficult to sustain cessation [43].

Facilitators in this domain included access to affordable and flexible cessation programs, which improved participation rates [45]. Additionally, technology-based interventions, such as mobile applications and online support platforms, were preferred by students, particularly those who smoked in social settings [46]. While these environmental factors were consistently reported, cultural and contextual differences across the countries where the included studies were conducted (the United States, Jordan, and Qatar) may influence how such barriers and facilitators operate, for example, through variations in tobacco policy and social norms.

#### 3.2.3. Social Influences

The presence of peer influence and social smoking norms was identified as a key barrier to cessation, as students who were surrounded by smokers found it more challenging to quit [44]. Additionally, a lack of social support from friends and family members discouraged quit attempts [47].

Facilitators in this domain included social accountability, where students who disclosed their quit attempts to supportive friends reported higher success rates [43]. Group-based interventions and peer support systems were also found to be effective in helping students stay motivated during the cessation process [45].

#### 3.2.4. Social/Professional Role and Identity

Barriers in this domain were influenced by cultural norms and gender expectations. Masculine cultural norms discouraged men from seeking professional help for smoking cessation, as quitting support was perceived as a sign of weakness [48]. For female students, on the other hand, fear of social stigma was a major barrier, particularly regarding how their smoking history might be perceived by family and in future personal relationships [48].

Facilitators included self-identifying as a non-smoker, which was associated with increased motivation to quit [44]. Additionally, older students were more likely to attempt cessation, possibly due to increased responsibilities and a shift in priorities [47].

#### 3.2.5. Beliefs About Capabilities

Self-efficacy was a major barrier in this domain, with many students overestimating their ability to quit without assistance, reducing the likelihood of seeking professional help [48]. Fear of relapse also discouraged quit attempts, particularly among students who had previously failed to quit [47]. Daily smokers were more likely to report low confidence in their ability to quit, further complicating cessation efforts [46].

Conversely, facilitators included previous quit attempts, which were associated with increased self-efficacy and a higher likelihood of cessation success [47]. Additionally, higher confidence in resisting smoking in social settings was linked to a greater likelihood of quitting [47].

#### 3.2.6. Knowledge

Barriers in this domain included a lack of awareness about nicotine addiction, with many students failing to recognize their dependence on nicotine [45]. Misconceptions about cessation methods were also common, as some students believed that nicotine replacement therapy was only meant for older smokers [45]. Underestimation of the health risks of occasional smoking further reduced motivation to quit [44].

Facilitators included education on nicotine withdrawal and cessation options, which was shown to increase motivation to quit [45]. Additionally, awareness of the immediate negative effects of smoking, such as shortness of breath and coughing, acted as a strong motivator for cessation [45].

#### 3.2.7. Intentions and Goals

Barriers in this domain included low internal motivation to quit, particularly among daily smokers who lacked a strong personal commitment to cessation [44]. Additionally, skepticism toward incentive-based programs discouraged students from participating in cessation initiatives [43].

Facilitators included higher readiness to quit, with students in more advanced stages of change demonstrating a stronger intention to quit smoking [47]. Moreover, autonomous motivation, where students quit for personal rather than external reasons, was positively associated with quit success [44].

#### 3.2.8. Beliefs About Consequences

Barriers in this domain included skepticism toward cessation services, with some students perceiving them as ineffective [48]. Additionally, beliefs that occasional smoking is not harmful were associated with delayed quit attempts [44].

Facilitators included increased awareness of the long-term health risks of smoking, which encouraged some students to attempt cessation [47]. Additionally, financial concerns, particularly the high cost of cigarettes, encouraged cessation efforts [45].

#### 3.2.9. Skills

Barriers in this domain included a lack of alternative stress management strategies, as many students relied on smoking to cope with academic and social pressures [45]. Additionally, low self-efficacy in handling social smoking situations made quitting particularly difficult [44].

Facilitators included higher self-efficacy, where students who believed they could resist smoking were more likely to maintain cessation [47]. Additionally, previous quit attempts helped students gain experience in managing withdrawal symptoms and avoiding smoking triggers [47].

#### 3.2.10. Reinforcement

Barriers in this domain included guilt associated with previous quit failures, which discouraged students from making another quit attempt [47].

Facilitators included financial incentives, such as gift cards, which motivated participation in cessation programs [43]. Additionally, the structured nature of incentive-based smoking cessation programs, such as the Quit and Win contest, provided external motivation to encourage smoking cessation [43].

#### 3.2.11. Emotion

Barriers in this domain included emotional dependence on smoking, as students often used smoking to manage stress, boredom, and social isolation [45]. Psychological factors, such as depression and fear of weight gain, also made quitting more challenging [47].

Facilitators included negative attitudes toward smoking, which were associated with increased motivation to quit, particularly among students who had seen their peers successfully quit [44]. Additionally, witnessing friends quit smoking served as emotional encouragement for those considering cessation [45]. The detailed barriers and facilitators to smoking cessation, categorized by the Theoretical Domains Framework, are presented in Table 2.

### 3.3. Identified Theoretical Frameworks

A total of five theoretical frameworks were identified across the studies included in this review. The identified frameworks were the Transtheoretical Model (TTM) [44,47], Problem Behavior Theory (PBT) [44], Health Belief Model (HBM) [48], Socio-Ecological Model (SEM) [48], and MAPS (Motivational and Problem-Solving) Counseling Framework [43]. Two studies did not report using a theoretical framework [45,46].

The TTM was used to assess readiness to quit smoking and understand the factors influencing smoking cessation behavior. Haddad and Petro-Nustas [47] applied TTM to evaluate its relevance for future smoking cessation programs, while Pinsker et al. [44] used it to analyze psychosocial determinants of cessation readiness among different subgroups of college student smokers.

The MAPS Counseling Framework was incorporated into the incentive-based Quit and Win contest to structure counseling sessions aimed at supporting smoking cessation [43].

The HBM was applied to explore individual-level barriers to seeking cessation services [48]. The model was applied to understand students’ perceptions of smoking-related risks, the benefits of quitting, and the barriers to cessation.

The SEM was employed to explore broader contextual influences on smoking cessation behaviors [48]. This framework demonstrated how individual, social, organizational, and cultural factors shape smoking cessation challenges and access to support services.

The PBT was applied to investigate the role of peer influence, personality traits, and behavioral engagement in smoking behaviors [44].

## 4. Discussion

Our review is the first theory-based scoping review to map the barriers and facilitators to smoking cessation among university students using the TDF. This allows for a systematic synthesis of the evidence and supports the development of more effective, theory-informed interventions. A total of 22 barriers and 20 facilitators were identified across multiple TDF domains from six studies conducted in three countries: the United States, Jordan, and Qatar, highlighting various individual, social, and environmental factors that influence smoking cessation efforts by university students. The findings suggest that smoking cessation among university students is shaped by a complex interaction of personal motivation, social influences, and access to cessation services.

Time constraints due to academic schedules [48] and financial limitations [45] were identified as barriers to smoking cessation, restricting students’ access to cessation services. These findings suggest that smoking cessation interventions should be offered to university students at no or low cost. One effective approach is the use of digital interventions; our findings indicate that digital tools were preferred by university students for smoking cessation [46]. Supported by previous research, digital interventions have been shown to be effective in supporting smoking cessation while remaining cost-efficient [49]. Additionally, access to flexible, low-cost cessation programs was also identified as a facilitator in this review [45]. Consistent with earlier studies, digital interventions may help overcome accessibility barriers by offering flexible, on-demand cessation support [50].

Cultural and gender norms also played a role in smoking cessation behaviors among university students. Masculine cultural expectations discouraged male students from seeking professional cessation support, while female students reported fear of social stigma associated with both smoking and quitting [48]. These findings underscore the importance of tailored interventions that address the specific cultural and gender-related barriers to smoking cessation within university settings. The need for culturally sensitive approaches is further supported by previous research [51].

Self-efficacy also played a role in smoking cessation behaviors among the population examined in this review. Many students overestimated their ability to quit without support, which reduced their likelihood of engaging in cessation interventions [48]. Daily smokers were particularly vulnerable, often reporting low confidence in their ability to quit [46]. In contrast, students who had made previous quit attempts demonstrated higher self-efficacy and a greater willingness to try quitting again [47].

These findings suggest that smoking cessation interventions should prioritize strategies aimed at enhancing self-efficacy. A promising approach involves the use of digital health tools, which have demonstrated positive effects on self-efficacy in smoking cessation [52], particularly when they incorporate gamified elements to engage users and reinforce progress through interactive and motivational features [53,54]. These gamified elements align with theoretical frameworks such as Self-Determination Theory, by satisfying psychological needs for autonomy, competence, and relatedness, and Self-Efficacy Theory, by enhancing users’ belief in their ability [55].

The influence of peer networks and social smoking norms was identified as a barrier in this review, making it difficult for students to quit in environments where smoking is socially accepted or encouraged [43]. Our findings indicate that students who frequently engaged in social smoking were less likely to seek cessation support [44]. However, consistent with previous research [56], peer support programs and group-based interventions were identified as facilitators [45], with students who disclosed their quit attempts to supportive peers reporting higher success rates [43]. Additionally, online social support, such as through social media platforms, may offer accessible and ongoing encouragement for smoking cessation among students who are surrounded by smokers [57]. Moreover, gamified interventions that incorporate social elements such as competition, collaboration, peer support, and teamwork have been shown to have a positive impact on behavior change [58,59].

### 4.1. Implications for Smoking Cessation Interventions

We found that several barriers to smoking cessation among university students were linked to environmental constraints, social influences, and psychological factors such as low self-efficacy. These findings highlight the need for tailored intervention strategies that are responsive to the unique challenges faced by this population. For instance, academic pressures and financial constraints were identified as barriers to accessing cessation services, suggesting that smoking cessation programs must be both flexible and affordable to meet students’ needs. Digital interventions were seen as effective tools in addressing these limitations, offering cost-effective, accessible, and on-demand support options that can be integrated into students’ daily routines.

Social smoking norms were also found to be a challenge, with students in smoking-supportive peer groups less likely to seek help or attempt to quit. This underscores the importance of peer-led and group-based interventions, which may be particularly effective in addressing social influences that normalize smoking, consistent with previous studies showing that peer-support interventions [60] and group-based programs [61] positively impact smoking cessation outcomes. Additionally, cultural and gender-related barriers, such as stigma associated with female smoking or unwillingness among male students to seek help due to masculine norms, point to the value of developing gender- and culturally sensitive programs. This finding aligns with evidence showing that culturally tailored interventions significantly improve quit rates compared to non-tailored approaches [51] and that gender-relevant programs can enhance cessation outcomes, particularly among women [62].

Considering these findings, smoking cessation efforts within university settings should prioritize the integration of low-cost, student-centered services into existing campus health systems. This includes offering nicotine replacement therapies at a reduced cost, incorporating smoking education into the curriculum, and leveraging digital platforms to engage students where they are. Peer support networks and social media communities, such as those on social media, could further enhance engagement, particularly for students who may feel isolated or lack in-person support. Tailored interventions that combine behavioral strategies, peer involvement, and accessible technology show strong potential for addressing the complex barriers to smoking cessation among university students.

### 4.2. Reflections on Theoretical Frameworks

This review identified five theoretical frameworks across the included studies, with the TTM being the most commonly applied [44,47]. In line with prior research, TTM has been widely utilized in smoking cessation studies and is recognized as a validated framework for understanding behavior change processes [63,64]. It has also guided the design of numerous cessation interventions aimed at improving quit rates and minimizing the likelihood of relapse during the quitting process [65,66].

The MAPS Counseling Framework was incorporated to structure counseling sessions aimed at supporting smoking cessation [43]. As MAPS is designed to be applicable across all stages of readiness to change [67], it is well-suited to addressing the varied motivational levels present among university student populations.

In one of the included studies [48], the HBM was used to examine individual-level perceptions of smoking risks, perceived benefits of quitting, and barriers to cessation, while the SEM was applied to explore broader contextual factors such as cultural norms, social support, and institutional environments. Such integration of individual-level models with broader frameworks, which has been emphasized in previous research [68], offers a comprehensive understanding of smoking cessation behaviors among university students by addressing both individual-level perceptions and broader contextual factors.

The PBT was applied in one of the included studies to examine how peer influence, personality traits, and behavioral engagement contribute to smoking behaviors among university students [44]. As a social-psychological framework, PBT provides valuable insights into the development and nature of problem behaviors [69] and has been used in understanding smoking behavior [70]. This understanding can ultimately support the design of tailored interventions that address both individual and social determinants of smoking.

While each framework contributes valuable insights into smoking cessation, they also present certain limitations when applied individually. For example, the TTM has been criticized for weak intervention guidance and questionable validity of stage assessments [71,72,73]. The HBM has been criticized for insufficient attention to social and cultural influences [74], while SEM’s multi-level approach, though comprehensive, may be difficult to operationalize due to its complexity [75]. Similarly, the MAPS framework requires trained counselors and face-to-face delivery [67], which may limit its scalability, and the PBT, while useful descriptively [76,77], offers limited guidance for intervention design.

Taken together, these frameworks highlight complementary elements of smoking cessation behavior, with some focusing primarily on individual-level cognitive and motivational processes such as TTM, while others emphasize broader social, cultural, and environmental determinants such as SEM. This suggests that an integrative, multi-level approach may provide a more comprehensive foundation for understanding and addressing smoking cessation among university students.

### 4.3. Strengths and Limitations

This review mapped the barriers and facilitators to smoking cessation among university students using the TDF, providing a structured and theory-driven approach to understanding the challenges faced by this population. The inclusion of both qualitative and quantitative studies enabled a comprehensive analysis of the factors influencing cessation. Furthermore, no restrictions were placed on publication date or geographical location, ensuring a broad and inclusive search strategy.

However, several limitations should be acknowledged. First, although the search strategy was not geographically restricted, all included studies were conducted in the United States, Jordan, and Qatar, which may limit the generalizability of the findings to university populations in other cultural or regional contexts. Second, most included studies relied on self-reported data, introducing the potential for recall bias and social desirability bias, which may have influenced the accuracy of reported smoking behaviors and cessation experiences. Third, consistent with the nature of scoping reviews, no formal quality assessment of the included studies was conducted, and therefore, the methodological rigor of the evidence remains uncertain. Fourth, the review focused exclusively on the experiences of university students and did not include the views of other relevant stakeholders such as university staff, faculty members, or healthcare providers. Fifth, our review and the included studies focused on cigarette smoking rather than on other forms of nicotine or tobacco use, such as e-cigarettes, which are often considered cessation aids rather than substitutes for cigarette smoking. Additionally, substances such as marijuana were not a focus of this review, as their use is less prevalent compared to cigarette smoking and is subject to legal restrictions in many countries. Sixth, we acknowledge that existing cessation interventions may have limitations that affect long-term cessation outcomes in this population. Although our review provides a foundation to support the development of more effective, theory-informed interventions, addressing these limitations was beyond the scope of this review and requires further investigation. Finally, the review was limited to studies published in English and Arabic, which may have led to the exclusion of relevant research published in other languages.

## 5. Conclusions

This scoping review is the first to apply the TDF to map the barriers and facilitators to smoking cessation among university students. Based on six studies conducted in the United States, Jordan, and Qatar, the review identified 22 barriers and 20 facilitators across multiple TDF domains, reflecting a complex interaction of individual, social, and environmental factors influencing smoking cessation behaviors in this population. While some findings, such as self-efficacy, mirror those reported in other TDF-based reviews [78], factors like academic stress appear more specific to the university setting [79,80], highlighting the importance of tailored interventions for university students.

Key findings indicate that time constraints, financial limitations, low self-efficacy, and social smoking norms are major barriers that hinder students’ access to cessation resources and motivation to quit. Conversely, flexible, low-cost digital interventions, peer support programs, and culturally sensitive, gender-responsive approaches were reported as facilitators. The use of gamified digital tools and online platforms appears particularly promising for increasing engagement and sustaining quit attempts.

Five theoretical frameworks were applied across the included studies, with the TTM being the most used. Other frameworks included the MAPS, HBM, SEM, and PBT. These models enhance the understanding of smoking cessation behaviors and inform the design of multi-level, theory-based interventions tailored to university student populations.

Our findings highlight the importance of tailoring smoking cessation strategies to meet the unique needs of university students. Interventions should be accessible, affordable, and embedded within campus health services, with a focus on integrating behavioral strategies, peer involvement, and accessible technology. Future research should explore the implementation and evaluation of such multifaceted interventions in diverse cultural contexts and consider the perspectives of university staff and healthcare providers to inform a whole-system approach.

Despite limitations related to geographic representation, self-reported data, and exclusion of the non-English/Arabic literature, this review offers a theory-driven foundation for developing and scaling effective smoking cessation interventions within university settings.

## Figures and Tables

**Figure 1 ijerph-22-00947-f001:**
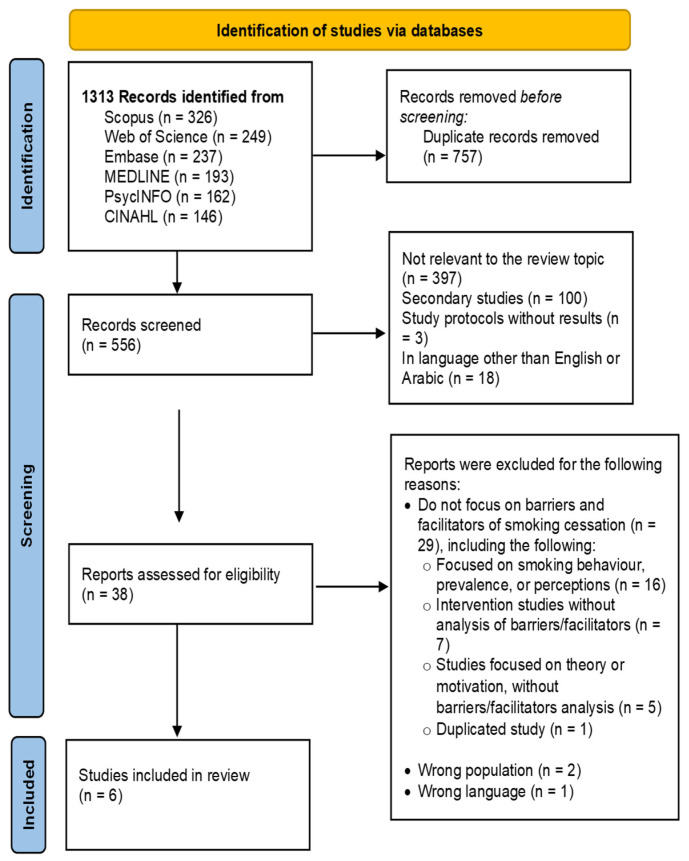
Study selection process. Source: Page MJ, et al. BMJ 2021; 372: n71. doi: 10.1136/bmj.n71. [42].

**Table 1 ijerph-22-00947-t001:** Characteristics of included studies.

Author/s Country	Aim	Research Design/ Data Collection Method	Sample Size/Population	Setting	Smoking Behavior	Key Findings
Thomas et al., 2015 [43] United States	To investigate social contingencies in smoking cessation	Qualitative focus groups; six focus groups with past Quit and Win participants, moderated using a semi-structured guide	27 university students, avg. age: 27.4, 63% female	Two midwestern universities in the U.S.	8.4 cigarettes/day, 40.7% nicotine dependent, 2.7/5 friends smoke	Social factors strongly influence quit success
Haddad and Petro-Nustas, 2006 [47] Jordan	To identify factors influencing students’ intention to quit smoking	Quantitative cross-sectional study; self-administered survey conducted in classrooms	800 university students, 90% male, avg. age: 20–22	Two public universities in Jordan	65% smoked ≥15 cigarettes/day, 84% considered quitting, 71% had attempted to quit before	Readiness was the strongest predictor of quitting, followed by past quit attempts and social support
Al-Jindi, Al-Sulaiman, and Al-Jayyousi, 2024 [48] Qatar	To explore barriers preventing students from accessing smoking cessation services	Qualitative cross-sectional study; semi-structured interviews with university students	20 university students, aged 18–30, mixed gender	University setting in Qatar	Majority were daily smokers, high nicotine dependence	Social and cultural factors strongly influence help-seeking behavior for smoking cessation
Pinsker et al., 2013 [44] United States	To examine psychosocial factors and substance use among daily and non-daily college student smokers and assess readiness to quit smoking	Quantitative cross-sectional study	4438 college students	Six colleges in the southeastern United States	10.2% daily smokers, 7.1% native non-daily smokers, 6.4% converted non-daily smokers	Converted non-daily smokers were more likely to be ready to quit than native non-daily smokers; readiness to quit associated with motivation, smoking frequency, and social smoking patterns
Staten and Ridner, 2007 [45] United States	To explore smoking cessation experiences among different groups of college students and their perspectives on cessation programs	Qualitative descriptive study; focus group interviews	19 18–24-year-old college students (former smokers, struggling smokers, and smokers with no intention to quit)	Large southeastern public university	Most students started smoking between ages 13 and 15, some struggled with addiction	College students need tailored cessation interventions that acknowledge their social environment and motivations
Berg et al., 2012 [46] United States	To examine differences in use of and interest in cessation strategies between daily and non-daily college student smokers	Quantitative cross-sectional study; online survey	800 undergraduate smokers, aged 18–25	A 4-year and a 2-year college in the Midwest, United States	65.8% non-daily smokers, 34.3% daily smokers	Non-daily smokers were less likely to seek assistance for quitting but equally interested in behavioral interventions compared to daily smokers

**Table 2 ijerph-22-00947-t002:** Barriers and facilitators to smoking cessation mapped to TDF domains.

TDF Domain (*n* = Total Studies for Barriers and Facilitators)	Barriers	Facilitators
Knowledge (*n* = 2)	Lack of awareness about nicotine addiction: many students did not recognize their dependence on nicotine [45]. Misinformation about cessation methods: some believed nicotine replacement therapy was only for older smokers [45]. Lack of awareness about the health risks of occasional smoking [44].	Education on nicotine withdrawal and cessation options increased motivation to quit [45]. Understanding the immediate negative effects of smoking (e.g., shortness of breath, coughing) encouraged quitting [45].
Skills (*n* = 3)	Lack of alternative stress management techniques: students used smoking as a coping mechanism [45]. Limited self-efficacy in handling social smoking situations [44].	Increased self-efficacy: students who believed they could resist smoking were more likely to quit [47]. Experience from previous quit attempts helped students understand their challenges and build skills [47].
Social/Professional Role and Identity (*n* = 3)	Masculine cultural norms discouraged help-seeking—men viewed quitting support as a sign of weakness [48]. Social stigma for women smokers—fear that families would find out or that smoking history would impact marriage prospects [48].	Having a strong self-identity as a non-smoker motivated some to quit [44]. Older students were more inclined to quit, possibly due to increased life responsibilities [47].
Beliefs about Capabilities (*n* = 3)	High self-efficacy in quitting without help: many students overestimated their ability to quit on their own [48]. Fear of relapse discouraged students from making quit attempts [47]. Low confidence in quitting, particularly among daily smokers [46].	Students with higher confidence in resisting smoking in social settings had a stronger intention to quit [47]. Previous quit attempts reinforced self-efficacy [47].
Beliefs about Consequences (*n* = 4)	Skepticism toward cessation services: some students believed they were ineffective [48]. Perception that occasional smoking is not harmful delayed quit attempts [44].	Recognition of the long-term health risks of smoking motivated students to quit [47]. Financial concerns (high cost of cigarettes) encouraged cessation [45].
Intentions and Goals (*n* = 3)	Low motivation and lack of internal drive to quit: more common among daily smokers [44]. Skepticism about contests and incentive programs [43].	High stage of readiness: students in advanced stages of the Transtheoretical Model were more likely to quit [47]. Autonomous motivation (quitting for personal reasons) predicted a stronger intention to quit [44].
Reinforcement (*n* = 2)	Past quit failures led to guilt and reluctance to try again [47].	Financial incentives (e.g., gift cards) encouraged participation in cessation programs [43]. The structure of the Quit and Win contest provided motivation [43].
Environmental Context and Resources (*n* = 4)	Social environment: being around other smokers increased relapse risk [43]. Time constraint: students had busy university schedules that prevented them from seeking cessation services [48]. Financial barriers: students found nicotine replacement therapies too expensive [45].	Access to flexible, low-cost cessation programs improved participation [45]. Digital interventions: students preferred web-based and mobile support programs [46].
Social Influences (*n* = 4)	Peer influence: social smoking made quitting more challenging [44]. Lack of social support: friends and family were often unsupportive [47].	Accountability through social support: having a support system increased quit success [43]. Group-based interventions and peer support were seen as effective strategies [45].
Emotion (*n* = 3)	Emotional dependence: students linked smoking to stress relief [45]. Psychological barriers (boredom, isolation, depression) made quitting harder [47].	Negative attitudes toward smoking increased readiness to quit [44]. Seeing friends successfully quit provided emotional encouragement [45].

## Data Availability

The original contributions presented in this study are included in the article/Appendix A. Further inquiries can be directed to the corresponding authors.

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
