# Peer review of "Barriers and Facilitators to Smoking Cessation Among University Students: A Scoping Review"

_ijerph, 2025, doi:10.3390/ijerph22060947_

Round 1

Reviewer 1 Report

Comments and Suggestions for Authors

Dear corresponding Author, thank you for submitting your work to IJERPH journal.

Brief Summary

This scoping review maps the existing literature on barriers and facilitators to smoking cessation among university students using the TDF. Six studies conducted in USA, Jordan and Qatar were included.

General Comments

The review adresses a relevant topic using TDF appropriately. The methodology is rigorus but the limited number of studies (six) and restricted geographical representation limit generalizability. There is no quality assessment of the included studies and a lack of comparative critical analysis of the identified theoretical frameworks.

Specific Comments

  • Lines 42-45: Missing specification of unique characteristics of university students that require different approaches. For example, you state that to develop effective smoking cessation interventions it is fundamental to understand the factors that influence cessation among this population, but you don't specify which unique caracteristics of university students (e.g. time constraints, specific social pressures) might require different approaches compared to general population. I would consider expanding this point to better highlight the need for research focused on this population.
  • Lines 62-65: The use of scoping review for this topic should be better justified. Why would it be more suitable in this case? It's not clear to me
  • Lines 79-81: Missing justification for choosing TDF over other frameworks. The TDF is certainly a valid tool, but a brief explanation of why it was chosen would strenghen the methodology
  • Figure 1: It would be useful to further categorize the 29 excluded studies, the criterion is not clear to me (e.g. how many focused only on smoking prevalence?).
  • Table 1: some reported studies have inconsistant format compared to others. In particular, Staten and Ridner's study (2007) has a different format than others, with the objective placed after the study design. This creates confusion when reading the table.
  • Lines 198-207: Insufficient attention to differences between university contexts in the three countries. I believe that specific cultural differences could significanty influence these environmental factors.
  • Lines 397-421: Missing critical analysis of integration between the different theoretical frameworks.
  • Lines 446-447: It would be useful to compare the results with other reviews using TDF in similar contexts, for example in different smoking cessation settings (but perhaps related), to highlight similarities or differences unique to university population.

In conclusion I believe the paper is of interest despite the limitations described, I think the authors can consider improving some weaknesses and critical issues and propose an updated version that I will read with interest, best regards

Reviewer 2 Report

Comments and Suggestions for Authors

The aim of this scoping review was to identify the barriers and facilitators of smoking cessation among university students. This is a very important public health issue. 

Research methodology is appropriate and transparently described. Results are described in detail. Discussion is sound, and conclusions provide directions of potential applications. 

What I missed and therefore want to suggest is adding a table briefly summarizing barriers and facilitators across the reported domains for improved readibility of results. 

Reviewer 3 Report

Comments and Suggestions for Authors

Manuscript review:  Barriers and Facilitators to Smoking Cessation Among University Students: A Scoping Review The authors have provided a scoping review of the barriers and facilitators to smoking cessation among university students. This study is notable and clearly fills a gap in the literature identifying factors that may influence the effectiveness of interventions needed to address smoking behavior among university students. This is a well-done study; the methodology for the review is thorough and the manuscript well written. I enjoyed reading the manuscript. My congratulations to the authors. Here are a few comments/suggestions: 

  1. The authors need to provide a sufficient background of the prevalence and effects of smoking among college students. The manuscript provides a general overview of this information but no specific information on university students. The authors can add this information to paragraph 2 or include an additional paragraph that provides sufficient context of this problem among students. 
  2. The authors have stated that their review is the "first theory-based scoping review" (See Line 312) but have not justified why this is important. I agree that this is a strength of the paper, but its significance needs to be highlighted to the reader. 

Reviewer 4 Report

Comments and Suggestions for Authors

This study does what it proposes to do. It articulates barriers and facilitators to tobacco/smoking cessation in college environments. The problem with this study is that it does not reference the following two major weaknesses in the section dealing with strengths and weaknesses toward the end of the paper.

  1. Separating “smoking” from use of other nicotine delivery products and marijuana use.

The introduction to the study uses the terms “smoking” and “tobacco use” as if these terms are synonymous. They are not. They do not reflect the fact that at least 20% of tobacco use in the general population consists of cigars, hookahs, e-cigarettes, oral tobacco and other products. Some of these products present risk of illness and death far lower than cigarettes.

The Introduction, limitations section or both should address the degree to which the studies reviewed and this paper separate smoking cessation from cessation of use of other tobacco-related non-pharmaceutical nicotine delivery products. It also should address whether marijuana use is confused with smoking, and whether the available studies simply do not separate such use.

This is not a trivial issue since the overall goal is prevention of cigarette-related illness and death, and some of these product present substantially less risk. This paper does not reflect the fact that switching from cigarettes to lower risk vape and oral products (ie Tobacco Harm Reduction) is considered by many to constitute “smoking cessation.”

All this will require a definition of “smoking cessation” as used in this paper in the Introduction plus a discussion of this issue in the Limitations section of the paper.

  1. Efficacy of cessation efforts.

Again, for the limitations section, there should be recognition that use of nicotine pharmaceuticals and other cessation modalities often offer only limited and short-term benefits, with smoking then being re-initiated. Thus, long-term reduction in smoking-related illness and death may require more than smoking-cessation programming.

Round 2

Reviewer 1 Report

Comments and Suggestions for Authors

I have carefully read the responses given by the authors to my concerns and suggestions, and I believe the effort has gone in the right direction. I think the article can be accepted in this form

Reviewer 4 Report

Comments and Suggestions for Authors

authors did a reasonable job in addressing my concerns.